# Adversarial Training and Provable Defenses: Bridging the Gap

**Mislav Balunović, Martin Vechev**
Department of Computer Science
ETH Zurich, Switzerland
{mislav.balunovic, martin.vechev}@inf.ethz.ch

## Abstract

We present COLT, a new method to train neural networks based on a novel combination of adversarial training and provable defenses. The key idea is to model neural network training as a procedure which includes *both*, the verifier and the adversary. In every iteration, the verifier aims to certify the network using convex relaxation while the adversary tries to find inputs inside that convex relaxation which cause verification to fail. We experimentally show that this training method, named convex layerwise adversarial training (COLT), is promising and achieves the best of both worlds – it produces a state-of-the-art neural network with certified robustness of 60.5% and accuracy of 78.4% on the challenging CIFAR-10 dataset with a 2/255 $L_\infty$ perturbation. This significantly improves over the best concurrent results of 54.0% certified robustness and 71.5% accuracy.

## 1 Introduction

The discovery of adversarial examples in deep learning (Szegedy et al., 2013; Biggio et al., 2013) has increased the importance of creating new training methods which produce accurate and robust neural networks with provable guarantees.

**Existing work: adversarial and provable defenses** Adversarial training (Goodfellow et al., 2015; Kurakin et al., 2017) provides a framework to augment the training procedure with adversarial inputs produced by an adversarial attack. Madry et al. (2018) instantiated adversarial training using a strong iterative adversary and showed that their approach can train models which are highly robust against the strongest known adversarial attacks such as Carlini & Wagner (2017). This method has also been able to train robust ImageNet models (Xie et al., 2019). While promising, the main drawback of the method is that when instantiated in practice, via an approximation of an otherwise intractable optimization problem, it provides no guarantees – it does not produce a certificate that there are no possible adversarial attacks which could potentially break the model. To address this lack of guarantees, recent line of work on provable defenses (Wong & Kolter, 2018; Raghunathan et al., 2018; Mirman et al., 2018) has proposed to train neural networks that are certifiably robust to a specific attacker threat model. However, these guarantees come at the cost of a significantly lower standard accuracy than models trained using adversarial training. This setting raises a natural question: can we leverage ideas from both, adversarial training techniques and provable defense methods, so to obtain models with high accuracy and certified robustness?

**This work: combining adversarial and provable defenses** In this work, we take a step towards addressing this challenge. We show that it is possible to train more accurate and provably robust neural networks using the *same* convex relaxations as those used in existing, state-of-the-art provable defense methods, but with a new, different optimization procedure inspired by adversarial training. Our optimization works as follows: (i) to certify a property (e.g., robustness) of the network, the verifier produces a convex relaxation of all possible intermediate vector outputs in the neural network, then (ii) an adversary now searches over this (intermediate) convex region in order to find, what we refer to as a *latent adversarial example* – a concrete intermediate input contained in the relaxation that when propagated through the network causes a misclassification which prevents verification, and finally (iii) the resulting latent adversarial examples are now incorporated into our training scheme

using adversarial training. Overall, we can see this method as bridging the gap between adversarial training and provable defenses (it can conceptually be instantiated with any convex relaxation). We experimentally show that the method is promising and results in a neural network with state-of-the-art 78.4% accuracy and 60.5% certified robustness on the challenging CIFAR-10 dataset with 2/255 $L_\infty$ perturbation (the best known existing results are 71.5% accuracy and 54.0% certified robustness from concurrent work of Zhang et al. (2020)).

**Main Contributions**   Our key contributions are:

- A new method, which we refer to as *convex layerwise adversarial training* (COLT), that can train provably robust neural networks and conceptually bridges the gap between adversarial training and existing provable defense methods.
- Instantiation of convex layerwise adversarial training using linear convex relaxations used in prior work (accomplished by introducing a projection operator).
- Experimental results showing convex layerwise adversarial training can train neural network models which achieve both, state-of-the-art accuracy and certified robustness on CIFAR-10 with 2/255 $L_\infty$ perturbation.
- Complete implementation of our training and certification methods in a system which we release at `https://github.com/eth-sri/colt`.

Overall, we believe the method presented in this work is a promising step towards training models that enjoy both, higher accuracy and higher certification guarantees. An interesting item for future work would be to explore instantiations of the method with other convex relaxations than the one considered here.

## 2   RELATED WORK

We now discuss some of the closely related work on robustness of neural networks.

**Heuristic adversarial defenses**   After the first introduction of adversarial examples (Szegedy et al., 2013; Biggio et al., 2013), defense mechanisms to train robust neural networks were built based on the inclusion of adversarial examples to the training set (Kurakin et al., 2017; Goodfellow et al., 2015). Models trained using adversarial training with projected gradient descent (PGD) (Madry et al., 2018) were shown to be robust against the strongest known attacks (Carlini & Wagner, 2017). This is in contrast to other defense mechanisms which have been broken by new attack techniques (Athalye et al., 2018). While models trained using adversarial training achieve robustness against strong adversaries, there are no guarantees that model is robust against any kind of adversarial attack under the threat model considered.

**Provable adversarial defenses**   There has also been considerable amount of work on methods to train classifiers with robustness guarantees. These approaches are typically based on Lipschitz regularization Hein & Andriushchenko (2017), linear (Wong & Kolter, 2018) or semidefinite (Raghunathan et al., 2018; Dvijotham et al., 2018a) relaxations, hybrid zonotope (Mirman et al., 2018) or interval bound propagation (Gowal et al., 2018). While these approaches obtain robustness guarantees, accuracy of these networks is relatively small and limits practical use of these methods.

There has also been recent work on certification of general neural networks, not necessarily trained in a special way. These methods are based on SMT solvers (Katz et al., 2017), abstract interpretation (Gehr et al., 2018), mixed-integer linear programs (Tjeng et al., 2019), linear relaxations (Weng et al., 2018; Zhang et al., 2018; Singh et al., 2019c) or combinations of those (Singh et al., 2019b;a).

Another line of work proposes to replace neural networks with a randomized classifier (Lecuyer et al., 2018; Cohen et al., 2019; Salman et al., 2019a) which comes with probabilistic guarantees on its robustness. While these approaches scale to larger datasets such as ImageNet (although with probabilistic instead of exact guarantees), their bounds come from the relationship between $L_2$ robustness and Gaussian distribution. In this paper, we consider general verification problem where input is not necessarily limited to an $L_p$ ball, but arbitrary convex set, as explained in Section 3.

## 3  BACKGROUND

In this work we consider a threat model where an adversary is allowed to transform an input $\boldsymbol{x} \in \mathbb{R}^{d_0}$ into any point from a convex set $\mathbb{S}_0(\boldsymbol{x}) \subseteq \mathbb{R}^{d_0}$. The set $\mathbb{S}_0$ can capture wide range of specifications such as $L_p$ perturbations (Wong & Kolter, 2018), geometric transformations (Balunović et al., 2019), semantic perturbations (Mohapatra et al., 2019) or camera imaging (Yang & Rinard, 2019). In the case of a threat model based on $L_\infty$ perturbations, which we experiment with later, the convex set will be defined as $\mathbb{S}_0(\boldsymbol{x}) = \{\boldsymbol{x}' \in \mathbb{R}^{d_0}, ||\boldsymbol{x} - \boldsymbol{x}'||_\infty < \epsilon\}$.

A neural network consisting of $k$ layers and parameters $\theta$ is represented as a function $h_\theta$ where $h_\theta = h_\theta^k \circ h_\theta^{k-1} \cdots \circ h_\theta^1$ and $h_\theta^i : \mathbb{R}^{d_{i-1}} \to \mathbb{R}^{d_i}$ denotes a transformation applied at hidden layer $i$. We also denote the function representing part of the neural network from layer $i$ to the final layer $k$ as $h_\theta^{i:k} = h_\theta^k \circ h_\theta^{k-1} \cdots \circ h_\theta^i$.

Our goal will be to prove a property on the output of the neural network, encoded via a linear constraint:
$$\boldsymbol{c}^T h_\theta(\boldsymbol{x}') + d < 0, \forall \boldsymbol{x}' \in \mathbb{S}_0(\boldsymbol{x}) \tag{1}$$
where $\boldsymbol{c}$ and $d$ are property specific vector and scalar values, respectively. This formulation is general enough to capture many interesting safety properties (Dvijotham et al., 2018b; Qin et al., 2019), including robustness to $L_p$ perturbations. The standard approach to train a model $h_\theta$ to satisfy this constraint is to define a surrogate loss $\mathcal{L}$ and solve the following min-max optimization problem:
$$\min_\theta \mathbb{E}_{(\mathrm{x,y}) \sim D} \max_{\boldsymbol{x}' \in \mathbb{S}_0(\boldsymbol{x})} \mathcal{L}(h_\theta(\boldsymbol{x}'), y) \tag{2}$$

Because inner maximization is intractable, most existing approaches replace it with an approximation. Depending on the used approximation, we distinguish two families of techniques.

**Adversarial training**  One family of methods, referred to as adversarial training (Goodfellow et al., 2015; Kurakin et al., 2017), replaces the maximum loss with a lower bound which is obtained using an adversarial attack. Madry et al. (2018) maximized the inner loss using a projected gradient descent (PGD) attack and found that, perhaps surprisingly, this optimization procedure succeeds in training deep architectures which are robust against the strongest known adversaries. While this provides strong empirical evidence that the resulting models are indeed robust, the approach offers no guarantees. Thus, it remains unclear whether there exist even stronger attacks that could break a model trained in this manner.

**Provable defenses**  A second family of methods to train certified neural networks is based on the computation of an upper bound to the inner loss, as opposed to a lower bound computed for adversarial training. These methods are typically referred to as provable defenses as they provide guarantees on the robustness of the resulting network, under any kind of attack inside the threat model. An upper bound is typically computed using linear relaxations (Wong & Kolter, 2018), interval propagation (Gowal et al., 2018) or methods combining interval bounds and linear relaxations (Mirman et al., 2018; Zhang et al., 2020). However, these methods suffer from two disadvantages.

First, due to the convex relaxations, an upper bound on the loss is typically not tight and can be quite loose. However, we believe this is less of an issue due to the fact that interval relaxations were shown to experimentally be able to train more provably robust models than methods based on linear relaxations (which usually produce tighter bounds than intervals). For example, Mirman et al. (2018); Zhang et al. (2020) report $\sim 28\%$ robust accuracy using pure interval training on CIFAR-10 with perturbation 8/255 while Wong et al. (2018) achieve 21% using linear relaxations.

Second, the way these methods construct the loss makes the relationship between the loss and the network parameters significantly more complex than in standard training. Given a network with $L$ layers and at most $N$ neurons in each layer, to compute the loss we have to perform $\mathcal{O}(LN^2)$ operations for interval training and $\mathcal{O}(LN^3)$ operations for linear relaxation based training. Intuitively, while there are exceptions, we can expect that a loss involving more mathematical operations is more difficult to minimize. We hypothesize that this complexity of loss computation (here quantified as number of operations) causes the resulting optimization problem for training the network to be more difficult, meaning these training methods often converge to a suboptimal solution. Our experimental results confirm this – we substantially outperform existing methods both in terms of accuracy and certified robustness using the *same* linear relaxation, but a different optimization procedure.

**Certification via convex relaxations**  We now formally describe how provable defenses perform certification. We denote the set of possible intermediate concrete vectors at layer $i$ that can be obtained by propagating vector $\boldsymbol{x}' \in \mathbb{S}_0(\boldsymbol{x})$ through the network as $\mathbb{S}_i(\boldsymbol{x}) = h_\theta^i(\mathbb{S}_{i-1}(\boldsymbol{x})) \subseteq \mathbb{R}^{d_i}$. As it is difficult to explicitly compute the set $\mathbb{S}_i(\boldsymbol{x})$, a standard approach is to approximate it via a convex relaxation $\mathbb{C}_i(\boldsymbol{x})$. As the input set is already convex, there is no need to introduce a relaxation, and thus we set $\mathbb{C}_0(\boldsymbol{x}) = \mathbb{S}_0(\boldsymbol{x})$. Given a neural network layer $h_\theta^i$ which transforms one set of vectors into another, we represent its corresponding convex relaxation transformer as $g_\theta^i$. That is, $g_\theta^i$ will transform one convex set into another convex set. More formally, for any set $\mathbb{D} \subseteq \mathbb{R}^{d_{i-1}}$, $g_\theta^i(\mathbb{D})$ is convex and $h_\theta^i(\mathbb{D}) \subseteq g_\theta^i(\mathbb{D})$. Then, we recursively define the effect of $g_\theta^i$ on a convex relaxation as $\mathbb{C}_i(\boldsymbol{x}) = g_\theta^i(\mathbb{C}_{i-1}(\boldsymbol{x})) \subseteq \mathbb{R}^{d_i}$. Finally, to certify robustness using the obtained convex relaxation, it is enough to check whether all output vectors in $\mathbb{C}_k(\boldsymbol{x})$ satisfy the linear constraint in Equation 1. If this is true, then all output vectors in $\mathbb{S}_k(\boldsymbol{x})$ satisfy the constraint as well due to the fact that $\mathbb{S}_k(\boldsymbol{x}) \subseteq \mathbb{C}_k(\boldsymbol{x})$.

## 4 Provable Defense via Convex Layerwise Adversarial Training

We now describe our convex layerwise adversarial training approach which yields a provable defense that bridges the gap between standard adversarial training and existing provable defenses.

**Motivation: latent adversarial examples**  Consider an already trained neural network model $h_\theta$ which we would like to certify using convex relaxations. A fundamental issue here is that certification methods based on convex relaxations can struggle to prove the target property (e.g., robustness) due to the iterative loss of precision introduced by the relaxation. More precisely, assume the neural network actually satisfies the property from Equation 1 for an input $\boldsymbol{x}$, meaning that $\boldsymbol{c}^T h_\theta(\boldsymbol{x}') + d < 0, \forall \boldsymbol{x}' \in \mathbb{S}_0(\boldsymbol{x})$. Naturally, this also implies that the neural network behaves correctly in the latent space of its first hidden layer in the region $\mathbb{S}_1(\boldsymbol{x})$. Formally, this means that $\boldsymbol{c}^T h_\theta^{2:k}(\boldsymbol{x}_1') + d < 0, \forall \boldsymbol{x}_1' \in \mathbb{S}_1(\boldsymbol{x})$. However, if one would use a certification method which replaces the region $\mathbb{S}_1(\boldsymbol{x})$ by its convex relaxation $\mathbb{C}_1(\boldsymbol{x})$, then it is possible that we would fail to certify our desired property. This is due to the fact that there may exist an input $\boldsymbol{x}_1' \in \mathbb{C}_1(\boldsymbol{x}) \setminus \mathbb{S}_1(\boldsymbol{x})$ such that $\boldsymbol{c}^T h_\theta^{2:k}(\boldsymbol{x}_1') + d \geq 0$. Of course, we could repeat the above thought experiment and possibly find more violating latent inputs in the set $\mathbb{C}_i(\boldsymbol{x}) \setminus \mathbb{S}_i(\boldsymbol{x})$ of any hidden layer $i$. The existence of points found in the difference between a convex relaxation and the true region is a fundamental reason for the failure of certification methods based on convex relaxations. For convenience, we refer to such points as *latent adversarial examples*. Next, we describe a method which trains the neural network in a way that aims to minimize the number of latent adversarial examples.

**Layerwise provable optimization via convex relaxations**  Our key observation is that the two families of defense methods described earlier are in fact different ends of the same spectrum: methods based on *adversarial training* maximize the cross-entropy loss in the first convex region $\mathbb{C}_0(\boldsymbol{x})$ while *provable defenses* maximize the same loss, but in the last convex region $\mathbb{C}_k(\boldsymbol{x})$. Both methods then backpropagate the loss through the network and update the parameters using SGD. However, as explained previously, certification methods may fail even before the last layer due to the presence of latent adversarial examples in the difference of the regions $\mathbb{C}_i(\boldsymbol{x})$ and $\mathbb{S}_i(\boldsymbol{x})$. A natural question then is – can we leverage adversarial training so to eliminate latent adversarial examples from hidden layers and obtain a provable network?

To this end, we propose adversarial training in layerwise fashion. The initial phase of training is equivalent to adversarial training as used by Madry et al. (2018). In this phase in the inner loop we repeatedly find an input in $\mathbb{C}_0(\boldsymbol{x})$ which maximizes the cross-entropy loss and update the parameters of the neural network so to minimize this loss using SGD. Note that the outcome of this phase is a model which is highly robust against strong multi-step adversaries. However, certification of this fact often fails due to the previously mentioned loss of precision in the particular convex relaxation being used, which then leads to the existence of latent adversarial examples in the hidden layers.

The next step of our training method is visually illustrated in Figure 1. Here, we propagate the initial convex region through the first layer of the network and obtain the convex region $\mathbb{C}_1(\boldsymbol{x})$. We then solve the optimization problem to find a *concrete point* $\boldsymbol{x}_1'$ inside of $\mathbb{C}_1(\boldsymbol{x})$ which produces the maximum loss when this point is propagated further through the network (this forward pass

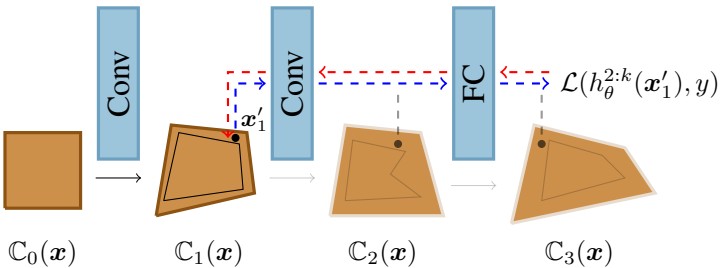

Figure 1: An iteration of convex layerwise adversarial training. Latent adversarial example $\boldsymbol{x}_1'$ is found in the convex region $\mathbb{C}_1(\boldsymbol{x})$ and propagated through the rest of the layers in a forward pass, shown with the blue line. During backward pass, gradients are propagated through the same layers, shown with the red line. Note that the first convolutional layer does not receive any gradients.

is shown with the blue line). Finally, we backpropagate the final loss (red line) and update the parameters of the network so to minimize the loss. Critically, we do not backpropagate through the convex relaxation in the first layer as standard provable defenses do (Wong & Kolter, 2018; Mirman et al., 2018; Gowal et al., 2018). We instead freeze the first layer and stop backpropagation after the update of the second layer. Because of this, our optimization problem is significantly easier – the neural network only has to learn to behave well on the *concrete points* that were found in the convex region $\mathbb{C}_l(\boldsymbol{x})$. This can be viewed as an extension of the robust optimization method that Madry et al. (2018) found to work well in practice.

We then proceed with the above process for later layers. Formally, this training process amounts to (approximately) solving the following min-max optimization problem at the $l$-th step:

$$\min_{\theta^{l+1:k}} \mathbb{E}_{(\mathrm{x,y})\sim D} \max_{\boldsymbol{x}_l'\in\mathbb{C}_l(\boldsymbol{x})} \mathcal{L}(h_\theta^{l+1:k}(\boldsymbol{x}_l'), y, \theta) \qquad (3)$$

Note that for $l = 0$ this formulation is equivalent to the standard min-max formulation in Equation 2 because $\mathbb{C}_0(\boldsymbol{x}) = \mathbb{S}_0(\boldsymbol{x})$. Our approach to solve this min-max optimization problem for every layer $l$ is shown in Algorithm 1. We initialize every batch by random sampling from the corresponding convex region. Then, in every iteration we use projected gradient descent (PGD) to maximize the inner loss in Equation 3. We first update $\boldsymbol{x}_j'$ in the direction of the gradient of the loss and then project it back to $\mathbb{C}_l(\boldsymbol{x}_j)$ using the projection operator $\Pi$. Note that this approach assumes an existence of an efficient projection method to the particular convex relaxation the method is instantiated with. In the next section, we show how to instantiate the training algorithm described above to a particular convex relaxation which is generally tighter than a hyperrectangle and where we derive an efficient projection operation.

---

**Algorithm 1:** Convex layerwise adversarial training via convex relaxations

> **Data:** $k$-layer network $h_\theta$, training set $(\mathcal{X}, \mathcal{Y})$, learning rate $\eta$, step size $\alpha$, inner steps $n$
> **Result:** Certifiably robust neural network $h_\theta$

1 **for** $l \leq k$ **do**
2      **for** $i \leq n_{epochs}$ **do**
3          Sample mini-batch $\{(\boldsymbol{x}_1, y_1), (\boldsymbol{x}_2, y_2), ..., (\boldsymbol{x}_b, y_b)\} \sim (\mathcal{X}, \mathcal{Y})$;
4          Compute convex relaxations $\mathbb{C}_l(\boldsymbol{x}_1), \mathbb{C}_l(\boldsymbol{x}_2), ..., \mathbb{C}_l(\boldsymbol{x}_b)$;
5          Initialize $\boldsymbol{x}_1' \sim \mathbb{C}_l(\boldsymbol{x}_1), \boldsymbol{x}_2' \sim \mathbb{C}_l(\boldsymbol{x}_2), ..., \boldsymbol{x}_b' \sim \mathbb{C}_l(\boldsymbol{x}_b)$;
6          **for** $j \leq b$ **do**
7              Update in parallel $n$ times: $\boldsymbol{x}_j' \leftarrow \Pi_{\mathbb{C}_l(\boldsymbol{x}_j)}(\boldsymbol{x}_j' + \alpha\nabla_{\boldsymbol{x}_j'}\mathcal{L}(h_\theta^{l+1:k}(\boldsymbol{x}_j'), y_j))$;
8          **end**
9          Update parameters $\theta \leftarrow \theta - \eta \cdot \frac{1}{b} \sum_{j=1}^b \nabla_\theta\mathcal{L}(h_\theta^{l+1:k}(\boldsymbol{x}_j'), y_j)$;
10      **end**
11      Freeze parameters $\theta_{l+1}$ of layer function $h_\theta^{l+1}$;
12 **end**

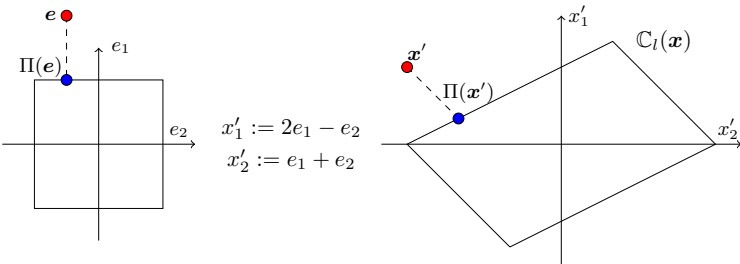

Figure 2: Projection to a region based on linear relaxation using change of variables.

## 5 CONVEX LAYERWISE ADVERSARIAL TRAINING USING LINEAR RELAXATIONS

So far we described the general approach of convex layerwise adversarial training. Now we show how to instantiate it for a particular convex relaxation based on linear approximations. If instead one would use interval approximation (Mirman et al., 2018; Gowal et al., 2018) as the convex relaxation, then all regions $\mathbb{C}_l(\boldsymbol{x})$ would be hyperrectangles and projection to these sets is fast and simple. However, the interval relaxation provides a coarse approximation which motivates the need to train with relaxations that provide tighter bounds. Thus, we consider linear relaxations which are generally tighter than those based on intervals.

In particular we leverage the same relaxation which was previously proposed in Wong & Kolter (2018); Weng et al. (2018); Singh et al. (2018) as an effective way to certify neural networks. Here, each convex region is represented as a set $\mathbb{C}_l(\boldsymbol{x}) = \{\boldsymbol{a}_l + \boldsymbol{A}_l\boldsymbol{e} \mid \boldsymbol{e} \in [-1, 1]^{m_l}\}$. Vector $\boldsymbol{a}_l$ represents the center of the set and the matrix $\boldsymbol{A}_l$ represents the affine transformation of the hypercube $[-1, 1]^{m_l}$. This representation is also known as zonotope abstraction (Ghorbal et al., 2009). The initial convex region $\mathbb{C}_0(\boldsymbol{x})$ is represented using $\boldsymbol{a}_0 = \boldsymbol{x}$ and $\boldsymbol{A}_0 = \epsilon\boldsymbol{I}_{d_0}$ is a diagonal matrix. Propagation of these convex regions through the network is out of the scope of this paper – a full description can be found in Wong & Kolter (2018) or Singh et al. (2018). At a high level, the convolutional and fully connected layers are handled by multiplying $\boldsymbol{A}_l$ and $\boldsymbol{a}_l$ by appropriate matrices. To handle the ReLU activation, we apply a convex relaxation that amounts to multiplying $\boldsymbol{A}_l$ and $\boldsymbol{a}_l$ by appropriately chosen diagonal matrices which depend on whether the ReLU is activated or not. Using this relaxation of ReLU, we recursively obtain all convex regions $\mathbb{C}_l(\boldsymbol{x})$. In practice, the term $\boldsymbol{A}_l\boldsymbol{e}$ can be computed without explicitly constructing matrix $\boldsymbol{A}_l$ because $\boldsymbol{A}_l\boldsymbol{e} = \boldsymbol{W}_l\boldsymbol{\Lambda}_{l-1}\boldsymbol{W}_{l-2}\cdots\boldsymbol{M}_0\boldsymbol{e}$. Due to the associativity of matrix multiplication, we can compute $\boldsymbol{A}_l\boldsymbol{e}$ by performing a chain of matrix-vector multiplications from right to left (instead of more expensive chain of matrix-matrix multiplications from left to right) and obtain vector $\boldsymbol{A}_l\boldsymbol{e}$. We provide more detailed description of this computation in Appendix A.

**Projection to linear convex regions**  To use our training method we now need to instantiate Algorithm 1 with a suitable projection operator $\Pi_{\mathbb{C}_l(\boldsymbol{x})}$. The key insight here is that the vector $\boldsymbol{x}' \in C_l(\boldsymbol{x})$ is uniquely determined by auxiliary vector $\boldsymbol{e} \in [-1, 1]^{m_l}$ where $\boldsymbol{x}' = \boldsymbol{a}_l + \boldsymbol{A}_l\boldsymbol{e}$. Then instead of directly solving for $\boldsymbol{x}'$ which requires projecting to $\mathbb{C}_l(\boldsymbol{x})$, we can solve for $\boldsymbol{e}$ instead which would uniquely determine $\boldsymbol{x}'$. Crucially, the domain of $\boldsymbol{e}$ is a hyperrectangle $[-1, 1]^{m_l}$ which is easy to project to. To visualize this further we provide an example in Figure 2. The goal is to project the red point $\boldsymbol{x}'$ in the right picture to the convex region $\mathbb{C}_l(\boldsymbol{x})$. To project, we first perform change of variables to substitute $\boldsymbol{x}'$ with $\boldsymbol{e}$ and then project $\boldsymbol{e}$ to the square $[-1, 1] \times [-1, 1]$ to obtain the blue point $\Pi(\boldsymbol{e})$ on the left. Then, we again perform change of variables to obtain the blue point $\Pi(\boldsymbol{x}')$ on the right, which is the projection of $\boldsymbol{x}'$ we were looking for.

Based on these observations, we modify Line 7 of Algorithm 1 to first update the coefficients $\boldsymbol{e}_j$ using the following update rule: $\boldsymbol{e}_j \leftarrow clip(\boldsymbol{e}_j + \alpha\boldsymbol{A}_l^T\nabla_{\boldsymbol{x}_j'}\mathcal{L}(\boldsymbol{x}_j', y_j), -1, 1)$. Here $clip$ is function which thresholds its argument between -1 and 1, formally $clip(t, -1, 1) = \min(\max(t, -1), 1)$. This is followed by an update to $\boldsymbol{x}_j'$ via $\boldsymbol{x}_j' \leftarrow \boldsymbol{a}_l + \boldsymbol{A}_l\boldsymbol{e}_j$, completing the update step.

**Efficient computation of convex regions**    While our representation of convex regions with matrix $\boldsymbol{A}_l$ and vector $\boldsymbol{a}_l$ has clean mathematical properties, in practice, a possible issue is that the matrix $\boldsymbol{A}_l$ can grow to be quite large. Because of this, propagating it through the network can be memory intensive and prohibit the use of larger batches. To overcome this difficulty, we propose two methods.

First method is based on the observation that $\boldsymbol{A}_l$ is quite sparse. We start with a very sparse, diagonal matrix $\boldsymbol{A}_0$ at the input. After each convolution, an element of matrix $\boldsymbol{A}_{l+1}$ is non-zero only if there is a non-zero element inside of its convolutional kernel in matrix $\boldsymbol{A}_l$. We can leverage this observation to precompute positions of all non-zero elements in matrix $\boldsymbol{A}_{l+1}$ and compute their values using matrix multiplication.

Second method is based on the idea from Wong et al. (2018). Their key insight is that convex regions $\mathbb{C}_l(\boldsymbol{x})$ can be computed approximately during training. They propose to use random projections from Li et al. (2007) to estimate lower and upper bound for each neuron. While they operate in dual framework, we apply the same approach in our primal view. In our experiments, we use this method as it is more efficient than working with sparse representation. After training, during certification, we compute regions $\mathbb{C}_l(\boldsymbol{x})$ exactly without the estimation.

This optimization is critical to enabling training to take place altogether. An interesting item for future work is further optimizing the current relaxation (via a specialized GPU implementation) or developing more memory friendly relaxations, so to scale the training to larger networks.

**Regularization**    One issue with the method presented in Section 4 is that there is no explicit mechanism that tries to make convex relaxation $\mathbb{C}_l(\boldsymbol{x})$ as close as possible to the exact region $\mathbb{S}_l(\boldsymbol{x})$ in order to avoid downstream loss of precision during later stages of the training, when layer $l$ is already frozen. To address this, we also incorporate additional regularizers similar to Xiao et al. (2019): $L_1$ regularization and ReLU stability regularization. The purpose of $L_1$ regularization is to learn sparse weight matrices and the goal of ReLU stability regularization is to have fewer crossing ReLU units, both of which are beneficial for precision. Our ReLU stability is different from Xiao et al. (2019) since we directly minimize the area induced by our linear convex relaxation. More details for these regularizers can be found in Appendix D.

## 6   CERTIFICATION OF NEURAL NETWORKS

After training a neural network via convex layerwise adversarial training, our goal is to certify the target property (e.g., robustness). Here we leverage several recent advances in certification techniques which are not fast enough to be incorporated into the training procedure, but which can significantly speed up the certification or increase its precision.

**Refinement of the linear approximation**    The linear relaxation of ReLU, which we are using, is parameterized by slopes $\boldsymbol{\lambda}$ of the linear relaxation. Prior work which employed this relaxation (Wong & Kolter, 2018; Weng et al., 2018; Singh et al., 2018) has chosen these slopes greedily by minimizing the area of the relaxation. During training we also choose $\boldsymbol{\lambda}$ in the same way. However, during certification, we can also optimize for the values of $\boldsymbol{\lambda}$ that give rise to the convex region inside of which the maximum loss is minimized. This optimization problem can be written as:

$$\min_{\boldsymbol{\lambda} \in [0,1]^{d_l}} \max_{\boldsymbol{x}' \in \mathbb{C}_l(\boldsymbol{x};\boldsymbol{\lambda})} \mathcal{L}(h_\theta^{l+1:k}(\boldsymbol{x}'), y)$$

Solving this is computationally too expensive inside the training loop, but during certification it is feasible to approximate the solution. We solve for $\boldsymbol{\lambda}$ using the Adam optimizer and clipping the elements between 0 and 1 after each update. We remark that the idea of learning the slope is similar to Dvijotham et al. (2018b) who propose to optimize dual variables in a dual formulation, however here we stay in the primal formulation.

**Combining convex relaxations with exact bound propagation**    During convex layerwise adversarial training we essentially train the network to be certified on all regions $\mathbb{C}_0(\boldsymbol{x}), ..., \mathbb{C}_k(\boldsymbol{x})$. While computing exact regions $\mathbb{S}_l(\boldsymbol{x}) \subseteq \mathbb{C}_l(\boldsymbol{x})$ is not feasible during training, we can afford it to some extent during certification. The idea is to first propagate the bounds using convex relaxations until

one of the hidden layers $l$ and obtain a region $\mathbb{C}_l(\boldsymbol{x})$. If training was successful, there should not exist a concrete point $\boldsymbol{x}'_l \in \mathbb{C}_l(\boldsymbol{x})$ which, if propagated through the network, violates the correctness property in Equation 1. We can encode both, the property and the propagation of the *exact bounds* $\mathbb{S}_l(\boldsymbol{x})$ using a Mixed-Integer Linear Programming (MILP) solver. Note that we can achieve this because we represent the region $\mathbb{C}_l(\boldsymbol{x})$ using a set of linear constraints, which may not be possible for general convex shapes. We perform the MILP encoding using the formulation from Tjeng et al. (2019). It is usually possible to encode only the last two layers using MILP due to the poor scalability of these solvers for realistic network sizes. One further improvement we also include is to tighten the convex regions $\mathbb{C}_l(\boldsymbol{x})$ using refinement via linear programming as described in Singh et al. (2019b). We remark that this combination of convex relaxation and exact bound propagation does not fall under the recently introduced convex barrier to certification Salman et al. (2019b).

## 7  EXPERIMENTAL EVALUATION

We now present an evaluation of our training method on the challenging CIFAR-10 dataset. All of our code, datasets, trained models and scripts to reproduce the experiments can be found at `https://github.com/eth-sri/colt`.

**Experimental setup**   We perform all experiments on a desktop PC using a single GeForce RTX 2080 Ti GPU and 16-core Intel(R) Core(TM) i9-9900K CPU @ 3.60GHz. We implemented training and certification in PyTorch (Paszke et al., 2017) and used Gurobi 9.0 as a MILP solver.

**Neural network architecture**   We evaluate on two architectures. First architecture is a 4-layer convolutional network: first 3 layers are convolutional layers with filter sizes 32, 32, 128, kernel sizes 3, 4, 4 and strides 1, 2, 2, respectively. Second architecture is a 3-layer convolutional network: first 2 layers are convolutional layers with filter sizes 32 and 128, kernel sizes 5 and 4, strides 2 and 2, respectively. In both architectures, convolutional layers are followed by a fully connected layer consisting of 250 hidden units. After each layer there is a ReLU activation. Final layer is a fully connected layer with 10 output neurons.

**Training**   During layerwise training we start with $\epsilon$ perturbation which is higher than the one we certify, and then decrease it by a certain factor when the training progresses to the next layer. In each stage of the training, we train for 200 epochs, starting from the same loss as in the previous stage and gradually annealing it to the loss of the current stage during first 60 epochs. We optimize using SGD with the initial learning rate 0.03 and after the initial 60 epochs we multiply the learning rate by 0.5 every 10 epochs. To find the best performing hyperparameters for training, we created a validation set consisting of random 5000 images from the training set and used it to tune the hyperparameters with SigOpt (Martinez-Cantin et al., 2018). We tuned batch size, initial $\epsilon$, factor to decrease $\epsilon$ after each layer, $L_1$ regularization and ReLU stability factors. All hyperparameter values used in our experiments are listed in Appendix C.1.

**Certification**   After training completes, we perform certification as follows: for every image, we first try to certify it using only linear relaxations (with the improvement of learned slopes, Section 6). If this fails, we encode the last layer as MILP and try again. Finally, if this fails we encode the ReLU activation after the last convolution using additional up to 50 binary variables and the rest using the triangle formulation Ehlers (2017). We consider an image to be not certifiable if we fail to certify it using these methods. We always certify the full test set of 10 000 images.

**Comparison to prior work**   We first train a robust network using our method for the $L_\infty$ perturbation 2/255. In this experiment, we used larger architecture with 3 convolutional and 1 fully connected layer. We perform convex layerwise adversarial training in 4 stages, for 200 epochs per stage, for a total of 800 epochs. The training takes 53, 164, 228, 250 seconds per epoch in the four respective stages of the training. It takes roughly 2 days to certify 10 000 images on a single GPU. Results are shown in Table 1. We always compare to the best reported and reproducible results in the literature on *any* architecture. We do not compare to smoothing-based approaches (Cohen et al., 2019), as these provide probabilistic instead of exact guarantees. Extensions to Cohen et al. (2019) such as Salman et al. (2019a) also use additional existing techniques such as pre-training

Table 1: Evaluation on CIFAR-10 dataset with $L_\infty$ perturbation 2/255

| Method | Accuracy(%) | Certified Robustness(%) |
|---|---|---|
| Our work | **78.4** | **60.5** |
| Zhang et al. (2020) | 71.5 | 54.0 |
| Wong et al. (2018) | 68.3 | 53.9 |
| Gowal et al. (2018) | 70.2 | 50.0 |
| Xiao et al. (2019) | 61.1 | 45.9 |
| Mirman et al. (2019) | 62.3 | 45.5 |

Table 2: Evaluation on CIFAR-10 dataset with $L_\infty$ perturbation 8/255

| Method | Accuracy(%) | Certified Robustness(%) |
|---|---|---|
| Our work | 51.7 | 27.5 |
| Zhang et al. (2020) | **54.5** | **30.5** |
| Mirman et al. (2019) | 46.2 | 27.2 |
| Wong et al. (2018) | 28.7 | 21.8 |
| Xiao et al. (2019) | 40.5 | 20.3 |

on ImageNet and unlabeled data which are orthogonal. We also do not compare to using cascades from Wong et al. (2018), as this improvement is also orthogonal to the method here. Thus, we only consider their best *single* network architecture (inline with prior work Zhang et al. (2020) which compares to a single architecture). We believe all methods listed in Table 1, including ours, would benefit from additional techniques such as cascades, pre-training and leveraging unlabeled data. Experimentally, we find that the neural network trained using our method substantially outperforms all existing approaches, both in terms of standard accuracy and certified robustness for 2/255. Note that here we are using the same linear relaxation as Wong et al. (2018), but our optimization procedure is different and shows significant improvements over the one used in their work. We also remark that concurrent work Zhang et al. (2020) reports 59.7% robustness against PGD which implies that even *empirical* robustness of their best model is lower than *certified* robustness of our network.

We also run the same experiment for $L_\infty$ perturbation 8/255 and present the results in Table 2. In this experiment, we used smaller architecture with 2 convolutional and 1 fully connected layer. Here we perform convex layerwise adversarial training in 3 stages, again for 200 epochs per stage, for a total of 600 epochs. The training takes 20, 67, 87 seconds per epoch in the three respective stages of the training. Here we do not include comparison with Gowal et al. (2018) as their results were found to be not reproducible (Mirman et al., 2019; Xu, 2019; Zhang et al., 2019), and the best reproducible results for this method can be found in Zhang et al. (2020). Here we substantially outperform all existing approaches except for the concurrent work of Zhang et al. (2020) whose method is based on a combination of interval and linear relaxation. We suspect that here the main issue is that our 3-layer network lacks capacity to solve this task, and capacity was found to be one of the key components necessary to obtain a robust classifier (Madry et al., 2018). Due to promising results for 2/255, we believe achieving state-of-the-art results for 8/255 is very likely an issue of instantiating our method with a convex relaxation that is more memory efficient, which we believe is an interesting item for future work.

**Analysis** Next, we analyze the effect of our convex layerwise adversarial training, also on CIFAR-10 with 2/255 and 8/255 $L_\infty$ perturbations. Recall that in the first stage, our training is equivalent to PGD from Madry et al. (2018) and in each of the successive stages, we freeze the current layer and retrain the rest of the network. In this experiment, we are interested in robustness of a model against latent adversarial attacks during each stage of the training. To perform this experiment, after each stage of the training we stored the intermediate model and ran latent adversarial attack on these models, on each of the layers. For latent adversarial attack, we perform PGD in the latent space, with 150 steps and step size of 0.01. Note that final models correspond to the models reported in Table 1 and Table 2.

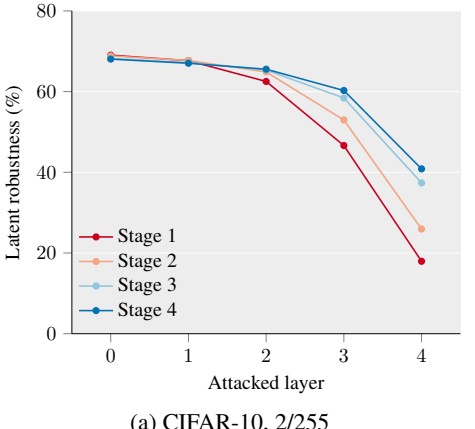 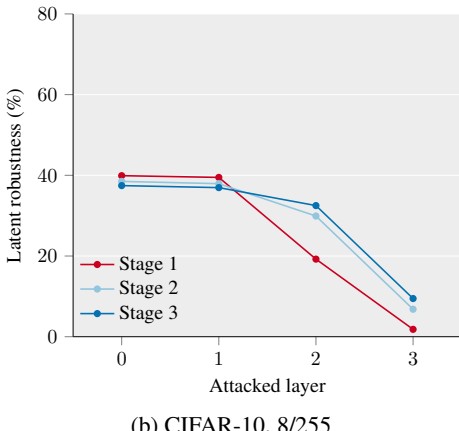

(a) CIFAR-10, 2/255  (b) CIFAR-10, 8/255

Figure 3: Effect of proposed convex layerwise adversarial training. After each stage of the training, we attack the model with a latent adversarial attack on each of the layers. Note that layer 0 represents standard PGD attack (attack in the input space).

The results are shown in Figure 3a and Figure 3b, for 2/255 and 8/255 perturbations, respectively. Each dot represents success rate of latent adversarial attack on a trained model, where each model is shown in a different color. Red line shows the model trained after the first stage, which is equivalent to PGD training from Madry et al. (2018). As expected, this model is robust against the standard adversarial attack in the input space which is denoted as attack on layer 0. However, this Stage 1 model lacks robustness in the deeper layers which prevents us from certifying the robustness using convex relaxations. Using convex layerwise adversarial training, the model progressively becomes robust to perturbations in the deeper layers. For example, Figure 3a shows that final model, after Stage 4, has 41% robustness in layer 4 which is significant improvement over Stage 1 model which has only 18% robustness in layer 4. Note that this means we can certify 41% using only linear relaxation, and 60.5% when learning the slopes and encoding part of the network as MILP, which is explained in Section 6. We observe similar results for 8/255 perturbation shown in Figure 3b.

**Other datasets** To further evaluate our method, we also experimented with other datasets: Street House View Numbers (SVHN) and MNIST Handwritten Digits. We report the full results in Appendix C.2 and give a brief summary here. On SVHN with $L_\infty$ perturbation 0.01 we also achieve state-of-the-art accuracy and certified robustness. On MNIST, we evaluated with perturbations 0.1 and 0.3. With $L_\infty$ perturbation 0.1 we achieve results comparable with best results from prior work, while with perturbation 0.3 our certified robustness is lower than the one achieved by approaches based on interval bound propagation (Zhang et al., 2020). We believe that, because of large perturbation of 0.3, random projections are imprecise and one would need to use the exact bounds which introduces much higher cost at runtime. This is also reflected in the poor performance of Wong et al. (2018) on this benchmark, as we use the same random projections as their work. We believe that instantiating our method with a convex relaxation that is more memory friendly than what we used would likely yield better results in this experiment.

## 8 CONCLUSION

We presented a new method to train certified neural networks. The key concept was to combine techniques from provable defenses using convex relaxations with those of adversarial training. Our method, named convex layerwise adversarial training (COLT), achieves state-of-the-art 78.4% accuracy and 60.5% certified robustness on CIFAR-10 with a 2/255 $L_\infty$ perturbation, significantly outperforming prior work when considering a single network (it also achieves competitive results on 8/255 $L_\infty$). The method is general and can be instantiated with *any* convex relaxation. A promising future work item is scaling to larger networks: this will require tight convex relaxations with a low memory footprint that allow for efficient projection.

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

Here we provide additional details that were omitted in the main body of the paper.

## A CONVEX RELAXATION USING LINEAR APPROXIMATIONS

In this section, we describe how to propagate convex relaxations of the form $\mathbb{C}_l(\boldsymbol{x}) = \{\boldsymbol{a}_l + \boldsymbol{A}_l\boldsymbol{e} \mid \boldsymbol{e} \in [-1,1]^{m_l}\}$ through the network, for a single input $\boldsymbol{x}$. As explained before, these relaxations were previously proposed in Wong & Kolter (2018); Weng et al. (2018); Singh et al. (2018). For the sake of completeness we describe them here using our notation. Depending on the form of function $h_\theta^i$ representing operation applied at layer $i$ we distinguish different cases. Here we assume that we have obtained region $\mathbb{C}_{i-1}(\boldsymbol{x})$ and our goal is to compute the region $\mathbb{C}_i(\boldsymbol{x})$ using convex relaxation $g_\theta^i$ of the function $h_\theta^i$.

**Initial convex region** Let $\epsilon$ be $L_\infty$ radius which we are certifying. First, we compute minimum and maximum pixel values for each pixel as $\boldsymbol{x}^l = \max(0, \boldsymbol{x} - \epsilon)$ and $\boldsymbol{x}^u = \min(1, \boldsymbol{x} + \epsilon)$. Then, we define initial convex region as:

$$m_0 := d_0$$
$$\boldsymbol{a}_0 := \frac{1}{2}(\boldsymbol{x}^l + \boldsymbol{x}^u)$$
$$\boldsymbol{A}_0 := \frac{1}{2}I_{d_0}(\boldsymbol{x}^u - \boldsymbol{x}^l)$$

**Convolutional and fully connected layers** For both convolutional and fully connected layers, the concrete update is given by $\boldsymbol{x}'_{i+1} = h_\theta^{i+1}(\boldsymbol{x}'_i) = \boldsymbol{W}_{i+1}\boldsymbol{x}'_i + \boldsymbol{b}_{i+1}$. Recall that $\boldsymbol{x}'_i = \boldsymbol{a}_i + \boldsymbol{A}_i\boldsymbol{e}$ where $\boldsymbol{e} \in [-1,1]^{m_i}$. We can then compute:

$$\begin{aligned} h_\theta^{i+1}(\boldsymbol{x}'_i) &= \boldsymbol{W}_{i+1}\boldsymbol{x}'_i + \boldsymbol{b}_{i+1} \\ &= \boldsymbol{W}_{i+1}(\boldsymbol{a}_i + \boldsymbol{A}_i\boldsymbol{e}) + \boldsymbol{b}_{i+1} \\ &= \boldsymbol{W}_{i+1}\boldsymbol{a}_i + \boldsymbol{b}_{i+1} + \boldsymbol{W}_{i+1}\boldsymbol{A}_i\boldsymbol{e} \end{aligned}$$

Using this formula, we define convex region $\mathbb{C}_{i+1}(\boldsymbol{x}) = \{\boldsymbol{a}_{i+1} + \boldsymbol{A}_{i+1}\boldsymbol{e} \mid \boldsymbol{e} \in [-1,1]^{m_{i+1}}\}$ where:

$$m_{i+1} := m_i$$
$$\boldsymbol{a}_{i+1} := \boldsymbol{W}_{i+1}\boldsymbol{a}_i + \boldsymbol{b}_{i+1}$$
$$\boldsymbol{A}_{i+1} := \boldsymbol{W}_{i+1}\boldsymbol{A}_i$$

**ReLU activation** In this case $h_\theta^{i+1}$ is ReLU activation function and $\boldsymbol{x}'_{i+1} = h_\theta^{i+1}(\boldsymbol{x}'_i) = \mathrm{ReLU}(\boldsymbol{x}'_i) = \max(\boldsymbol{x}'_i, 0)$ applied componentwise. We will explain the transformation of a single element $x'_{i,j} = a_{i,j} + \boldsymbol{A}_{i,j}\boldsymbol{e}$. We first compute lower bound $l_{i,j}$ and upper bound $u_{i,j}$ of element $x'_{i,j}$ in the set $\mathbb{C}_i(\boldsymbol{x})$:

$$l_{i,j} = a_{i,j} - \sum_{k=1}^{m_i} |A_{i,j,k}|$$
$$u_{i,j} = a_{i,j} + \sum_{k=1}^{m_i} |A_{i,j,k}|$$

If $u_{i,j} < 0$ then $\mathrm{ReLU}(x'_{i,j}) = 0$ and if $l_{i,j} > 0$ then $\mathrm{ReLU}(x'_{i,j}) = x'_{i,j}$. In the other case where $0$ is between $l_{i,j}$ and $u_{i,j}$ we define $\mathrm{ReLU}(x'_{i,j}) = \lambda_{i,j}x'_{i,j} + \mu_{i,j}e_{m_i+j}$ where $e_{m_i+j} \in [-1,1]$ is a coefficient for a new error term. Formulas for $\lambda_{i,j}$ and $\mu_{i,j}$ are the following:

$$\lambda_{i,j} = \frac{u_{i,j}}{u_{i,j} - l_{i,j}}$$
$$\mu_{i,j} = \frac{-u_{i,j}l_{i,j}}{2(u_{i,j} - l_{i,j})}$$

This computation can also be written in the matrix form as $\text{ReLU}(\boldsymbol{x}_i') = \boldsymbol{\Lambda}_{i+1}\boldsymbol{x}_i' + \boldsymbol{M}_{i+1}\boldsymbol{e}_{new}$ where $\boldsymbol{\Lambda}_{i+1}$ and $\boldsymbol{M}_{i+1}$ are diagonal matrices with elements computed as above. More formally, $\boldsymbol{\Lambda}_{i+1}$ and $\boldsymbol{M}_{i+1}$ are both $d_i \times d_i$ diagonal matrices with diagonal entries $\Lambda_{i+1,j,j} = \lambda_{i,j}$ and $M_{i+1,j,j} = \mu_{i,j}$.

We can rewrite $\boldsymbol{x}_{i+1}' = \boldsymbol{\Lambda}_{i+1}\boldsymbol{x}_i' + \boldsymbol{M}_{i+1}\boldsymbol{e}_{new} = \boldsymbol{\Lambda}_{i+1}(\boldsymbol{a}_i + \boldsymbol{A}_i\boldsymbol{e}) + \boldsymbol{M}_{i+1}\boldsymbol{e}_{new}$. Finally, new convex region $\mathbb{C}_{i+1}(\boldsymbol{x}) = \{\boldsymbol{a}_{i+1} + \boldsymbol{A}_{i+1}\boldsymbol{e} \mid \boldsymbol{e} \in [-1,1]^{m_{i+1}}\}$ is defined as:

$$m_{i+1} = m_i + d_i$$
$$\boldsymbol{a}_{i+1} = \boldsymbol{\Lambda}_{i+1}\boldsymbol{a}_i$$
$$\boldsymbol{A}_{i+1} = [\boldsymbol{\Lambda}_{i+1}\boldsymbol{A}_i, \boldsymbol{M}_{i+1}]$$

where [] denotes concatenation of matrices along the column dimension.

## B  STATISTICAL ESTIMATION OF BOUNDS USING RANDOM PROJECTIONS

Here we describe how we apply random projection approach from Wong et al. (2018) to estimate the bounds during training. While Wong et al. (2018) operate in dual framework, their method to statistically estimate the bounds during training can also be applied in primal framework which we use in this paper. Recall that lower and upper bound for each neuron are computed as

$$l_{i,j} = a_{i,j} - \sum_{k=1}^{m_i} |A_{i,j,k}| \tag{4}$$

$$u_{i,j} = a_{i,j} + \sum_{k=1}^{m_i} |A_{i,j,k}| \tag{5}$$

Thus, we need to compute $||\boldsymbol{A}_i||_1$, which is $L_1$ norm of each row in the matrix $\boldsymbol{A}_i$. Wong et al. (2018) propose a method, based on the result from Li et al. (2007), to estimate $||\boldsymbol{A}_i||_1$ using Cauchy random projections. Here $\boldsymbol{A}_i$ is a matrix with $d_i$ rows and $m_i$ columns, where $d_i$ is dimensionality of the output vector in the $i$-th layer and $m_i$ is number of unit terms in the definition of region $\mathbb{C}_i(\boldsymbol{x})$. The method of random projections samples standard Cauchy random matrix $\boldsymbol{R}$ of dimensions $m_i \times k$ ($k$ is number of projections) and then estimates $||\boldsymbol{A}_i||_1 \approx \text{median}(|A_i\boldsymbol{R}|)$. First, we expand $\boldsymbol{A}_i$ as:

$$\begin{aligned}
\boldsymbol{A}_i &= [\boldsymbol{W}_i\boldsymbol{A}_{i-1}] \\
&= [\boldsymbol{W}_i[\boldsymbol{\Lambda}_{i-1}\boldsymbol{A}_{i-2}, \boldsymbol{M}_{i-1}]] \\
&= [\boldsymbol{W}_i\boldsymbol{\Lambda}_{i-1}\boldsymbol{A}_{i-2}, \boldsymbol{W}_i\boldsymbol{M}_{i-1}] \\
&= [\boldsymbol{W}_i\boldsymbol{\Lambda}_{i-1}\boldsymbol{W}_{i-2}\boldsymbol{A}_{i-3}, \boldsymbol{W}_i\boldsymbol{M}_{i-1}] \\
&= [\boldsymbol{W}_i\boldsymbol{\Lambda}_{i-1}\boldsymbol{W}_{i-2}[\boldsymbol{\Lambda}_{i-3}\boldsymbol{A}_{i-4}, \boldsymbol{M}_{i-3}], \boldsymbol{W}_i\boldsymbol{M}_{i-1}] \\
&= [\boldsymbol{W}_i\boldsymbol{\Lambda}_{i-1}\boldsymbol{W}_{i-2}\boldsymbol{\Lambda}_{i-3}\boldsymbol{A}_{i-4}, \boldsymbol{W}_i\boldsymbol{\Lambda}_{i-1}\boldsymbol{W}_{i-2}\boldsymbol{M}_{i-3}, \boldsymbol{W}_i\boldsymbol{M}_{i-1}] \\
&= \cdots \\
&= [\boldsymbol{W}_i\boldsymbol{\Lambda}_{i-1}\boldsymbol{W}_{i-2}\boldsymbol{\Lambda}_{i-3}\cdots\boldsymbol{M}_0, \ldots, \boldsymbol{W}_i\boldsymbol{\Lambda}_{i-1}\boldsymbol{W}_{i-2}\boldsymbol{M}_{i-3}, \boldsymbol{W}_i\boldsymbol{M}_{i-1}]
\end{aligned}$$

In the formula above, we set $\boldsymbol{M}_0 = \boldsymbol{A}_0$. Now,

$$||\boldsymbol{A}_i||_1 = ||\boldsymbol{W}_i\boldsymbol{\Lambda}_{i-1}\boldsymbol{W}_{i-2}\boldsymbol{\Lambda}_{i-3}\cdots\boldsymbol{M}_0||_1 + \cdots + ||\boldsymbol{W}_i\boldsymbol{\Lambda}_{i-1}\boldsymbol{W}_{i-2}\boldsymbol{M}_{i-3}||_1 + ||\boldsymbol{W}_i\boldsymbol{M}_{i-1}||_1$$

To calculate $\boldsymbol{A}_i\boldsymbol{R}$ we split $\boldsymbol{R} = [\boldsymbol{R}_0, \boldsymbol{R}_2, ..., \boldsymbol{R}_{i-1}]$ and compute:

$$\boldsymbol{A}_i\boldsymbol{R} = \boldsymbol{W}_i\boldsymbol{\Lambda}_{i-1}\boldsymbol{W}_{i-2}\boldsymbol{\Lambda}_{i-3}\cdots\boldsymbol{M}_0\boldsymbol{R}_0 + \boldsymbol{\Lambda}_{i-1}\boldsymbol{W}_{i-2}\boldsymbol{M}_{i-3}\boldsymbol{R}_{i-3} + \cdots + \boldsymbol{W}_i\boldsymbol{M}_{i-1}\boldsymbol{R}_{i-1}$$

Crucially, each summand can now be efficiently computed due to the associativity of matrix multiplication by performing the multiplication from right ot left. Finally, we compute the elementwise absolute value of $\boldsymbol{A}_i\boldsymbol{R}$ and take the median of each row, resulting in estimate of $||\boldsymbol{A}_i||_1$. Using this approach, we obtain statistical estimate of bounds $l_{i,j}, u_{i,j}$ in Equation 4.

## C  ADDITIONAL EXPERIMENTAL RESULTS AND HYPERPARAMETERS

In this section we present additional results on SVHN and MNIST datasets.

## C.1 HYPERPARAMETERS

Here we list all hyperparameters used in our experiments.

Table 3: Hyperparameter values for CIFAR-10 with $L_\infty$ perturbation 2/255

| Hyperparameter | Value |
|---|---|
| $L_1$ regularizer | 1e-5 |
| $\epsilon_{start}$ | $1.05 \cdot (2/255)$ |
| $\epsilon_{inc}$ | 1.1 |
| batch size | 100 |
| ReLU stability regularizer | 0.005 |
| ReLU stability increase factor | 1.5 |

Table 4: Hyperparameter values for CIFAR-10 with $L_\infty$ perturbation 8/255

| Hyperparameter | Value |
|---|---|
| $L_1$ regularizer | 5e-6 |
| $\epsilon_{start}$ | $1.05 \cdot (8/255)$ |
| $\epsilon_{inc}$ | 1.2 |
| batch size | 150 |
| ReLU stability regularizer | 0.003 |
| ReLU stability increase factor | 1.5 |

**CIFAR-10** For the experiments on CIFAR-10 dataset, we show hyperparameter values for $L_\infty$ perturbation 2/255 in Table 3 and for 8/255 in Table 4. Across both experiments, we use 8 steps during the latent adversarial attack and each step size is 0.25 (where perturbations are normalized between -1 and 1). In both cases, we use initial learning rate 0.03 and multiply learning rate by 0.5 every 10 epochs, after the initial 60 epochs. During the first 60 epochs, we mix the loss from the previous layer and the current layer (with current layer factor increasing linearly) and train each stage for total of 200 epochs. The rest of the hyperparameters are tuned on the validation set using Martinez-Cantin et al. (2018) for each experiment separately and shown in Table 3 and Table 4.

Table 5: Hyperparameter values for SVHN with $L_\infty$ perturbation 0.01

| Hyperparameter | Value |
|---|---|
| $L_1$ regularizer | 5e-5 |
| $\epsilon_{start}$ | 0.01 |
| $\epsilon_{inc}$ | 1.2 |
| batch size | 200 |
| ReLU stability regularizer | 0.005 |
| ReLU stability increase factor | 1.5 |

**SVHN** For the experiments on SVHN dataset, we use 40 steps for the latent adversarial attack with step size 0.035. We train using Adam (Kingma & Ba, 2014) with initial learning rate 0.0001 which is multiplied by 0.5 every 20 epochs, after the initial 60 epochs. Same as for CIFAR-10, we mix the losses during the initial 60 epochs and train each stage for a total of 200 epochs. We did not tune the hyperparameters for this experiment. The hyperparameters are shown in Table 5.

**MNIST** For the experiments on MNIST dataset, we use 40 steps for the latent adversarial attack with step size 0.035. We train using Adam (Kingma & Ba, 2014) with initial learning rate 0.0001 which is multiplied by 0.5 every 10 epochs, after the initial mixing epochs. When training with perturbation 0.1 we use 100 mixing and 300 total epochs, while for perturbation 0.3 we use 60 mixing and 150 total epochs for each stage of the training. We did not tune the hyperparameters for this experiment. The hyperparameters are shown in Table 6 and Table 7.

Table 6: Hyperparameter values for MNIST with $L_\infty$ perturbation 0.1

| Hyperparameter | Value |
| --- | --- |
| $L_1$ regularizer | 1e-5 |
| $\epsilon_{start}$ | 1.5 |
| $\epsilon_{inc}$ | 1.0 |
| batch size | 100 |
| ReLU stability regularizer | 0.005 |
| ReLU stability increase factor | 1.5 |

Table 7: Hyperparameter values for MNIST with $L_\infty$ perturbation 0.3

| Hyperparameter | Value |
| --- | --- |
| $L_1$ regularizer | 1e-5 |
| $\epsilon_{start}$ | 0.3 |
| $\epsilon_{inc}$ | 1.1 |
| batch size | 100 |
| ReLU stability regularizer | 0.0005 |
| ReLU stability increase factor | 1.0 |

## C.2 EXPERIMENTAL EVALUATION ON SVHN AND MNIST

We also evaluated our method on SVHN and MNIST datasets.

**SVHN** We evaluated our method on SVHN dataset. For this experiment, we used convolutional network with 2 convolutional layers of kernel size 4 and stride 1 with 32 and 64 filters respectively. These convolutional layers are followed by a fully connected layer with 100 neurons. Each of the layers is followed by a ReLU activation function. Results of this experiment are shown in Table 8. We certified full SVHN test dataset consisting of 26 031 images. Our network has both, higher accuracy and higher certified robustness, than networks trained using prior approaches.

**MNIST** In the next experiment, we evaluated on MNIST dataset. For this experiment, we used convolutional network with 2 convolutional layers with kernel sizes 5 and 4, and strides 2 followed by 1 fully connected layer. Each of the layers is followed by a ReLU activation function. We certified the full test set, consisting of 10 000 images.

For perturbation 0.1, convolutional layers have filter sizes 32 and 64, and fully connected layer has 100 neurons. Results are presented in Table 9. Here, our numbers are comparable to those of state-of-the-art approaches.

For perturbation 0.3, convolutional layers have filter sizes 32 and 128, and fully connected layer has 250 neurons. Results are presented in Table 10. Here, our certified robustness is lower than state-of-the-art. We believe this is due to the imprecise estimates of lower and upper bound via random projections. This is also reflected in relatively poor performance of Wong et al. (2018) who also rely on the same statistical estimates. Thus, for MNIST dataset and perturbation 0.3 it is likely necessary to use exact propagation instead of the estimates. However, as this also induces large cost to the runtime, the most promising approach may be a new convex relaxation which is more memory efficient.

## D REGULARIZATION

During training, we use two regularization mechanisms to make our convex relaxation tighter, both previously proposed in Xiao et al. (2019).

First, we use $L_1$-norm regularization which is known to induce sparsity in the weights of the network. Xiao et al. (2019) has shown that weight sparsity helps to induce more stable ReLU units which in turn makes our convex relaxation tighter (as for stable ReLU units it is already precise).

Table 8: Evaluation on SVHN dataset with $L_\infty$ perturbation 0.01

| Method | Accuracy (%) | Certified robustness (%) |
|---|---|---|
| Our work | **88.5** | **70.2** |
| Gowal et al. (2018) | 85.2 | 62.4 |
| Wong & Kolter (2018) | 79.6 | 59.3 |
| Dvijotham et al. (2018a) | 83.4 | 62.4 |

Table 9: Evaluation on MNIST dataset with $L_\infty$ perturbation 0.1

| Method | Accuracy (%) | Certified robustness (%) |
|---|---|---|
| Our work | **99.2** | 97.1 |
| Gowal et al. (2018) | 98.9 | **97.7** |
| Zhang et al. (2019) | 99.0 | 94.4 |
| Wong et al. (2018) | 98.9 | 96.3 |
| Dvijotham et al. (2018a) | 98.8 | 95.6 |
| Mirman et al. (2019) | 98.7 | 95.8 |
| Xiao et al. (2019) | 99.0 | 95.6 |

Second, in the $i$-th phase of training, we explicitly introduce a loss based on the volume of convex region $C_{i+1}$. To make the relaxation tighter and minimize the volume, for each neuron $j$ in layer $i + 1$ we add a loss of the form $\max(0, -l_{i+1,j}) \max(0, u_{i+1,j})$. This loss corresponds to the area under ReLU relaxation, and derivation of this formula can be found in e.g. Singh et al. (2018) for a derivation.

Table 10: Evaluation on MNIST dataset with $L_\infty$ perturbation 0.3

| Method | Accuracy (%) | Certified robustness (%) |
|---|---|---|
| Our work | 97.3 | 85.7 |
| Gowal et al. (2018) | **98.3** | 91.9 |
| Zhang et al. (2020) | 98.2 | **93.0** |
| Wong et al. (2018) | 85.1 | 56.9 |
| Mirman et al. (2019) | 96.6 | 89.3 |
| Xiao et al. (2019) | 97.3 | 80.7 |

