# OpenReview forum: "Adversarial Training and Provable Defenses: Bridging the Gap"
_ICLR.cc/2020/Conference — Accept (Talk)_

### Official Review · AnonReviewer1 · 2019-10-19
**Official Blind Review #1**

**Rating:** 8

**Review:**

Summary: the paper introduces a novel protocol for training neural networks that aims at leveraging the empirical benefits of adversarial training while allowing to certify the robustness of the network using the convex relation approach introduced by Wong & Kolter. The key ingredient is a novel algorithm for layer-wise adversarial (re-)training via convex relaxations. On CIFAR-10, the proposed protocol yields new state-of-the-art performance for certifying robustness against L_inf perturbations less than 2/255, and comparable performance over existing methods for perturbations less than 8/255 (where the comparison excludes randomized-smoothing based approaches as proposed by Cohen et al.).

The proposed methodology seems original and novel. The concept of latent adversarial examples, the layer-wise provable optimization techniques and the sparse representation trick are interesting in their own regard and could be valuable ingredients for future work in this direction. The improvement over the state-of-the-art on CIFAR-10 for perturbations less than 2/255 is significant (although I wouldn't call it substantial). For perturbations less than 8/255 the picture is less clear. The authors' explanation that they couldn't achieve state-of-the-art certified robustness because of smaller network capacity makes sense, however, it also highlights that their protocol doesn't scale as well as previous approaches.

I am not concerned about the missing comparison with randomized smoothing-based approaches (I find the rationale provided in Section 2 convincing).

The discussion of the relatively weak performance of previous provable defenses on page 3 is a bit vague, e.g. the statement that "the way these methods construct the loss makes the relationship between the loss and the network parameters significantly more complex than in standard training", thus causing the "resulting optimization problem to be more difficult". To me, these are one and the same thing, and a bit more rigour in the argumentation would be advisable here, in my opinion.

-------------

I acknowledge I have read the authors' response and also the other reviews/comments which confirm my opinion that this paper is worthy to be published at ICLR.

**Experience Assessment:**

I have read many papers in this area.

**Review Assessment: Checking Correctness Of Derivations And Theory:**

I did not assess the derivations or theory.

**Review Assessment: Checking Correctness Of Experiments:**

I assessed the sensibility of the experiments.

**Review Assessment: Thoroughness In Paper Reading:**

I made a quick assessment of this paper.

---

> ### Author Response · Authors · 2019-11-13
> **Response to Reviewer #1**
>
> Thank you for your feedback. Below we answer the main concerns.
>
> Q: Could you discuss your claims about relatively weak performance of previous provable defenses more rigorously?
>
> → Yes, we now clarify what exactly we meant there. Note that we offer only intuition and have no theoretical proof of the claim itself. When we say that the relationship between the loss function and parameters is more complex, this can be quantified in terms of the number of mathematical operations. For instance, propagating through a linear layer with N inputs and N outputs requires O(N^2) operations for standard training, O(N^2) operations for interval training and O(N^3) for training with linear relaxations from prior work. Our next sentence refers to the difficulty of minimizing this loss function. Intuitively, while there are exceptions, we can expect that a loss which involves more mathematical operations to be more difficult to minimize.
>
> Q: The authors' explanation that they couldn't achieve state-of-the-art certified robustness because of smaller network capacity makes sense, however, it also highlights that their protocol doesn't scale as well as previous approaches.
>
> → We have now managed to scale our approach to larger networks using the approach of Wong et al. (2018) to statistically estimate bounds during training. Please see our main points above for a more detailed description.

---

> > ### Comment · AnonReviewer1 · 2019-11-15
> > **Thank you for the response.**
> >
> > It's great to hear that you have managed to scale your approach to larger networks now. This certainly further strengthens your contribution.
> >
> > I'm also satisfied with the clarification regarding the relatively weak performance of previous provable defences.

---

### Official Review · AnonReviewer3 · 2019-10-19
**Official Blind Review #3**

**Rating:** 6

**Review:**


Summary:
This paper provides a promising new general training methodology to obtain provably robust neural networks (towards adversarial input perturbations). The paper provides promising experimental results on CIFAR-10 by obtaining state-of-the-art certified accuracy while also simultaneously improving clean accuracy. The paper is overall well-written and the algorithm is clearly described.

Important questions to be answered:
I find the need to clarify my understanding and request for more information in order to make a decision.

--Methodology/motivation for the method: I am trying to understand abstractly what the proposed layerwise training is trying to optimize. To be concrete, let's compare to the relaxation of Wong and Kolter (which this paper uses in the instantiation of layerwise adversarial training). What's the exact difference?

One way to view this is the following: The same training objective, but a different way to optimize. The new proposal to train involves freezing weights until one layer iteratively starting from the input layer. It is possible that this kind of training provides some inductive bias in finding better solutions. Is this an appropriate understanding?
However, the paper’s experimental results unfortunately change the certification procedure. In other words, they haven’t evaluated the same training objective as that of Wong and Kolter. Hence, it’s not clear if the gains are from the better networks, or better certification method, or network being better suited for certification by the method used. The phrase “same relaxation” is not appropriately used. Their certification procedure uses a different (and tighter) relaxation.


--Effect on latent adversarial examples: I am unable to understand why this training procedure would reduce the number of latent adversarial examples. The definition of latent adversarial examples seems to suggest that it’s the gap between the actual set of activations corresponding to the input perturbations and the convex hull. However, the proposed layerwise adversarial training procedure involves replacing the actual set S_i with the convex hull C_i when freezing things till below i-1. I do not follow how the proposed method tries to make C_i = S_i. Implicitly the optimization objective does try to make C_i small because the bounds being optimized are tighter when C_i is small. But this is true even for normal certified training, and not sure what changes in the new training procedure.


Specific experimental results that would help:
--Certified accuracy on using the same LP based certification procedure used in Wong and Kolter with the new layerwise trained networks
--The paper’s own certification procedure (a combination of previous methods) on the network from Wong and Kolter or a note on why that doesn’t apply (if it doesn’t)
--The paper currently provides only one data point to suggest this training method is superior. Would be good to try SVHN or MNIST. MNIST is perhaps “essentially” solved for small \eps. But would be good to see if the training method offers gains at larger \eps. In general, would be good to see more consistent gains.
--The paper reports results on first 1000 examples of CIFAR10 test set. From my personal experience, there is a lot of variability in the robustness of test examples when evaluated on 1000 random test instances. Especially since the paper doesn't take a random subset, it might be good to make sure the gains are consistent on some other subset. The Wong et al. baseline is evaluated on the entire test set for example, and hence might not be a fair comparison? What's the Wong et al. accuracy on just the first 1000 test exampes

Overall, I am leaning towards accept but need some conceptual and empirical clarification from the authors (detailed above).


**Experience Assessment:**

I have published one or two papers in this area.

**Review Assessment: Checking Correctness Of Derivations And Theory:**

I carefully checked the derivations and theory.

**Review Assessment: Checking Correctness Of Experiments:**

I carefully checked the experiments.

**Review Assessment: Thoroughness In Paper Reading:**

I read the paper thoroughly.

---

> ### Author Response · Authors · 2019-11-13
> **Response to Reviewer #3**
>
> Thank you for your feedback. Below we answer the main concerns.
>
> Q: Why do you change the certification procedure to one that is different from what was used in training? Can you apply your certification procedure on the networks from Wong & Kolter?
>
> → This is because during certification we can afford to use a more expensive certification method based on a complete verifier. We believe this approach, leveraging complete verifiers after training, is now standard and was already applied by Gowal et al. and Xiao et al.
>
> Other works have in fact already certified some of the networks from Wong et al. using complete verifiers. These results can be found in the works of Tjeng et al. and Salman et al. The certified robustness obtained using these verifiers is indeed slightly higher than the one in Wong et al. However, the best result of Wong et al., presented in our Table 1, is achieved with a residual network which is too large to apply our certification procedure.
>
> Q: I do not follow how the proposed method tries to make C_i = S_i. Could you elaborate the effect your training method has on latent adversarial examples?
>
> → We will try to clarify this point here. We also fixed a typo in Line 11 of Algorithm 1 (we freeze layer l + 1 instead of l) which might have caused some misunderstanding.
>
> First, we would like to clarify the definition of latent adversarial examples which you characterized as “the gap between the actual set of activations corresponding to the input perturbations and the convex hull”. This is actually not correct. It is possible that for a network there is a (large) gap between the actual set and the convex relaxation and at the same time no latent adversarial example exists. This would happen if the network behaves correctly on the entire difference between the convex relaxation and the true region. Training networks to behave this way is precisely the aim of our method.
>
> At the i-th stage of training, layers 1, 2, …, i of the network are frozen. This means that both regions S_i and C_i are fixed (which means our training is not trying to make S_i = C_i). Now, our training method performs two steps. First, it tries to find a latent adversarial example in the difference between C_i and S_i. Second, it updates the non-frozen parameters of the network so that the loss induced by the latent adversarial example is minimized. After enough such updates, this loss will decrease enough so that no latent adversarial example is remaining.
>
> Finally, as mentioned earlier in the main points, we remark that we found a regularizer similar to Xiao et al. useful to increase the performance. This regularizer is indeed trying to make C_i = S_i by inducing a loss on the volume of region C_i. However, our training itself has a different goal, as explained above.
>
> Q: The paper currently provides only one data point to suggest this training method is superior. Would be good to try SVHN or MNIST. MNIST is perhaps “essentially” solved for small \eps. But would be good to see if the training method offers gains at larger \eps. In general, would be good to see more consistent gains.
>
> → To further evaluate the performance of our method we evaluated our method on MNIST and SVHN datasets and included the results in Appendix C. Please see the main response for the summary.
>
> Q: Could there be significant variability in results due to the fact that only 1000 images from the test set were certified?
>
> → To check the amount of variability, we certified another random subset of 1000 images, with little difference in the results. Please see the main response for the results of this experiment.

---

### Official Review · AnonReviewer4 · 2019-10-31
**Official Blind Review #4**

**Rating:** 8

**Review:**

This paper was very clearly written and easy to follow. Kudos to the authors. In particular, the experimental evaluation section was exceptionally clear. Thanks to the authors for making the paper so easy to review. The “Main Contributions” section was excellent as well as it allows the reader to quickly understand what the paper is claiming.

The introduction & related work section was very clear, and seemed to quickly get the reader up to speed.

Minor critiques:

- It’s not clear to me that the network size is actually as impressive an improvement as is implied. Barring an extensive hyper-parameter search that demonstrated that this network architecture is the smallest possible that could achieve the presented results, I strongly suspect that applying techniques from papers like EfficientNet [1] or MobileNet would allow the authors of Mirman et. al (2018) to reduce the number of parameters required to achieve their results. I don’t think this takes away from the paper, though- the results are strong despite that. I would encourage the authors to weaken the claims that the only better network is 5 times larger.

- In general, I would have liked to see more evaluation- e.g. I would have liked to see more results with a variety of perturbations (2 through 8, not just 2 & 8), and on a variety of datasets.

Questions to the authors:

- How robust is the algorithm to architecture choice?
- What happens if you change the test set? E.g. instead of evaluating on the first 1000 images, what if you evaluate on another random subset? Does that make a difference? I’m concerned that the subset of the test set the authors are using for evaluation isn’t representative of the entire test set.
- Is the current architecture the largest network that can be run? I would be interested in seeing how network size affects the performance of your technique.
- What hyper-parameter tuning did you do? What other network architectures did you try?
- How do the comparison methods compare in terms of training time/machines used? E.g. do all the methods reported in Table 1 use similar amounts of computing power?


Overall, this is a great paper, with some interesting results presented in a tight, clear manner. While I would like to see more experiments on larger datasets- e.g. ImageNet- the results seem solid and absolutely worthy of publication.

[1]: https://arxiv.org/abs/1905.11946v2
[2]: https://arxiv.org/abs/1704.04861

**Experience Assessment:**

I have published one or two papers in this area.

**Review Assessment: Checking Correctness Of Derivations And Theory:**

I assessed the sensibility of the derivations and theory.

**Review Assessment: Checking Correctness Of Experiments:**

I carefully checked the experiments.

**Review Assessment: Thoroughness In Paper Reading:**

I read the paper thoroughly.

---

> ### Author Response · Authors · 2019-11-13
> **Response to Reviewer #4**
>
> Thank you for your feedback. Below we answer the main concerns.
>
> Q: It’s not clear to me that the network size is actually as impressive an improvement as is implied. I strongly suspect that applying techniques from papers like EfficientNet [1] or MobileNet would allow the authors of Mirman et. al (2018) to reduce the number of parameters required to achieve their results. I would encourage the authors to weaken the claims that the only better network is 5 times larger.
>
> → We agree that approaches from prior work could benefit from the techniques you mentioned to further reduce sizes of their networks. Our claim was referring only to the networks that were reported in the respective papers. However, as we have now also scaled our approach to larger networks, we modified the sentence and no longer claim that better networks are at least 5 times larger.
>
> Q: In general, I would have liked to see more evaluation- e.g. I would have liked to see more results with a variety of perturbations (2 through 8, not just 2 & 8), and on a variety of datasets.
>
> → To further evaluate the performance of our method we evaluated on MNIST and SVHN datasets and included results in Appendix C. Please see the main response in the summary. We chose perturbation values 2 and 8 because other values were not evaluated in prior work so we could not compare. Due to time constraints, we were unable to evaluate other perturbation values, however we will do so for the next version.
>
> Q: What hyper-parameter tuning did you do? What other network architectures did you try?
>
> → For training hyper-parameters we chose number of steps and step size of PGD to be the same as in Madry et al. (2018). We also chose L_1 regularization factor the same as Xiao et al. We always used batch size 50. The only two parameters we tuned were the epsilon used for training and the factor for ReLU stability regularization. As training is relatively costly, we experimented with a few different values and chose the one which minimizes adversarial loss in the final layer, on the training set.
>
> Q: Is the current architecture the largest network that can be run? I would be interested in seeing how network size affects the performance of your technique.
>
> → As mentioned before, we managed to scale our approach to larger networks. Compared to the smaller network used at submission time, we improved the accuracy by 4% and certified robustness by 2.2% on CIFAR-10 with 2/255 perturbation. For the final revision we will evaluate the method on a wider range of architectures, however at the moment this was not possible due to the limited rebuttal time period.
>
> Q:  How do the comparison methods compare in terms of training time/machines used? E.g. do all the methods reported in Table 1 use similar amounts of computing power?
>
> → Methods in Table 1 are very different in terms of computing power. While it is hard to directly compare them, Gowal et al. report their method takes 3.5 seconds per epoch and Wong et al. takes 2 minutes per epoch on the MNIST dataset. On MNIST, our method takes 2.5 minutes per epoch while training the first layer, 5 minutes per epoch for the second layer and 10 minutes per epoch for the third layer. Our CIFAR-10 networks take roughly 1 day to train on 1 GeForce RTX 2080 Ti GPU.

---

> > ### Comment · AnonReviewer4 · 2019-11-15
> > **Response to author response (reviewer #4)**
> >
> > Thank you for your response. That answered my questions. I would encourage the information about hyper-parameters and computing power, to be included in the paper. Other than that, I think this is a solid contribution to the literature, and am excited to see future work by the authors.

---

### Public Comment · ~Anthony_Wittmer1 · 2019-09-28
**Questions about the claim**

Hi, it is a great work but I have some questions about some claim in this paper.

In this paper, the claim about "We do not compare to smoothing-based approaches Cohen et al. (2019)[1], as these provide probabilistic instead of exact guarantees." may not be true.

For an input image, the model of Randomized Smoothing can give a robustness radius R, where for any perturbations $\left \| \delta \right \|_2 \leq R$, the model can provide the robustness guarantee. So I do not know why the authors make the above claim.

Moreover, Cohen et al. have compared their smoothing method with the method of Wong et al. (2018), which is a baseline in this paper. So I think it necessary to compare to smoothing-based approaches, .Cohen et al. (2019) and [2]. In addition, randomized smoothing is currently the only approach that can provide provable robustness  guarantees on ImageNet-scale problems.

The idea of combining the adversarial training and the provable defense is not vey novel. Previous work [2] has combimed the adversarial training and the provable defense (randomized smoothing) to boost the provable robustness.


[1] Certified Adversarial Robustness via Randomized Smoothing. ICML 2019
[2] Provably Robust Deep Learning via Adversarially Trained Smoothed Classifiers. upcoming NeurIPS 2019

---

> ### Public Comment · ~Jeremy_Cohen1 · 2019-09-29
> **different setttings**
>
> Hi Anthony,
>
> (I don't know the authors of this submission, I just came across your comment.)   There are two aspects of randomized smoothing that are unsatisfying:
>    (1) We don't certify the robustness of a neural network, we certify the robustness of a smoothed neural network g, which is a (deterministic) classifier whose predictions cannot be evaluated exactly, only approximated to arbitrarily high confidence.  Alternatively, you could view the Monte Carlo approximation to the smoothed neural network (i.e. a classifier which returns the majority vote of the base classifier over 1000 randomly corrupted inputs) as a randomized classifier g_hat, and you could say that we "probabilistically" certify the robustness of this classifier g_hat, in that we give guarantees of the form: for every input x+delta in a ball around x of radius R, 99% of the time when you evaluate g_hat at (x+delta) you would see g_hat(x+delta) .= cA.
>
> (2) Our certification procedure for certifying the robustness of g around x is also probabilistic, in the sense that there is always some probability that it will "fail", by returning a radius larger than the radius in which g is truly robust.  In our paper, we set this failure probability so low that there is absolutely no doubt that the true certified accuracy of g is more than a hair away from the "approximate" certified accuracies that we reported in the paper.  But it is still sort of unsatisfying to not be able to deterministically certify the robustness of the smoothed classifier.
>
> Given these disadvantages of randomized smoothing, I certainly think that research on the problem of certifying neural network classifiers is worthwhile in its own right, even though the numbers don't currently match those of randomized smoothing.

---

> > ### Author Response · Authors · 2019-10-04
> > **Thank you for helpful clarifications**
> >
> > Dear Jeremy, thank you for your helpful clarifications of the limitations of randomized smoothing and differences with certification of neural network classifiers. We certainly agree that both research directions are worth pursuing.
> >
> > The authors

---

> > ### Comment · Area_Chair1 · 2019-10-23
> > **Follow-on**
> >
> > I would also add that randomized smoothing has another disadvantage on inference time - since at inference time a randomized smoothing classifier has to average predictions of several random perturbations of the input data, it significantly slows down inference, which can be a deal-breaker in latency critical applications (like for models that power web search, Google translate etc.)

---

> ### Author Response · Authors · 2019-10-04
> **Response to main concerns**
>
> Dear Anthony,
>
> Thanks for your interest in our work. Below we respond to your main concerns:
>
> Q: “In this paper, the claim about "We do not compare to smoothing-based approaches Cohen et al. (2019)[1], as these provide probabilistic instead of exact guarantees." may not be true. For an input image, the model of Randomized Smoothing can give a robustness radius R, where for any perturbations $\left \| \delta \right \|_2 \leq R$, the model can provide the robustness guarantee. So I do not know why the authors make the above claim. “
>
> → Please see Jeremy’s response on limitations of randomized smoothing. In this work, we certify neural network classifiers with *exact* guarantees, while randomized smoothing certifies smoothed neural network classifier with probabilistic guarantees. While we believe both approaches are interesting, they are not fully comparable.
>
> Q: “The idea of combining the adversarial training and the provable defense is not vey novel. Previous work [2] has combimed the adversarial training and the provable defense (randomized smoothing) to boost the provable robustness.”
>
> → In this work, we propose the combination of adversarial training and certification of neural networks with exact guarantees. To the best of our knowledge, such a combination was not considered in prior work. The work you mention uses adversarial training to improve smoothed classifier, which ultimately provides stronger *probabilistic* guarantees for the same smoothed classifier. As mentioned in the previous question, in light of this, we believe our combination is novel. We will further clarify the differences with the smoothing based approach.
>
> The authors

---

### Public Comment · ~Greg_Yang1 · 2019-11-08
**A Suggestion for the Abstract**

Dear Authors,

Thanks for an interesting paper. While not comparing to randomized smoothing approaches in the paper is fine, perhaps it is misleading to claim, unconditionally, state-of-the-art accuracies in the abstract. For one, I believe a typical reader would assume the randomized smoothing is compared in the paper based on the abstract. Another reason is that the numbers reported is very far from the unconditional state-of-the-art, which achieves 68% provable accuracy and 87.2% clean accuracy (via SmoothAdv + ImageNet pretraining) [1].

I think some simple conditionals here would suffice, such as "nonprobabilistic certificates with no extra data" or something of the sort.

I apologize in advance for such an "annoying" comment, but I hope this would only improve the presentation of your paper from the get-go (in the abstract)!

[1] https://arxiv.org/abs/1906.04584

---

> ### Author Response · Authors · 2019-11-13
> **Response to the concern**
>
> Thank you for your feedback. We believe it is enough to modify our claim that we achieve a “model with state-of-the-art accuracy and certified robustness” to “state-of-the-art neural network”. A smoothed classifier is not a neural network (e.g. this is explained in related work in [1] and the comment by Jeremy below), so we believe this clarification should help address your comment. We updated the abstract in the newly updated PDF to reflect this.

---

### Author Response · Authors · 2019-11-13
**Improved results and answers to common questions**

We thank the reviewers for their comments. We first explain the improvements we introduced since the time of submission and then proceed to answer the questions raised by the reviewers.

## Improved results

Since the time of submission we scaled our approach to larger networks and improved the results. On CIFAR-10, these improvements led to training a network with 78.8% standard accuracy (4% improvement) and 58.1% (2.2% improvement) certified robustness for 2/255 perturbations. For 8/255 perturbation, we trained a network with 49.7% (3.5% improvement) standard accuracy and 26.0% (1.6% improvement) certified robustness. We updated the paper with new network and results.

We list the changes we incorporated to achieve these improvements:

- We applied random projections from Wong et al. (2018) to statistically estimate the region C_l instead of computing it exactly. We note that the general method still stays the same, so sections 3, 4 and 5 were not changed (except one clarification sentence in section 5), but this instantiation allows it to scale to larger networks. In Appendix B, we provide full derivation of the bounds using random projections. During certification, we also still use the same procedure as before (with exact guarantees, without estimating the bounds).

- In the projection operator, we avoid the computation of full matrix A_l and instead only compute matrix-vector product A_l * e by changing the order of computation, which we clarified now in Section 5.

- Similarly as Xiao et al., we incorporated additional regularizer to introduce ReLU stability. Our regularizer is slightly different as it is tailored to minimize the volume for the particular linear convex relaxation we are using. The regularizer is explained in Appendix D.

## Common questions:

R3, R4: Could there be significant variability in results due to the fact that only 1000 images from the test set were certified?

→ First, we clarify that we always evaluate standard accuracy on the full test set. To check the amount of variability in the certification results, we evaluated our CIFAR-10 network with 2/255 perturbation on another random subset consisting of 1000 images. On this subset we can certify 56.7% images, compared to 58.1% on the first 1000 images. In the next revision of our paper, we will evaluate all 10 000 images to match the evaluation setting used by prior work. Given the repetition experiment, we believe the results will not change significantly.

R3, R4: Could you add more evaluation to see the performance on a variety of datasets and perturbations?

→Yes. We performed additional experiments on SVHN and MNIST datasets and provided the results in Appendix C.
For SVHN and perturbation 0.01 we also achieve state-of-the-art standard accuracy and certified robustness.
For MNIST and perturbation 0.1 our results are comparable to state-of-the-art, while for perturbation 0.3 our certified accuracy is lower than the one achieved by approaches based on interval propagation. We believe that, because of large perturbation of 0.3, random projections are imprecise and one would need to use the exact bounds which introduces much higher cost at runtime. This is also reflected in the poor performance of Wong et al. (2018) who use the same random projections. We believe that instantiating our method with a convex relaxation that is more memory friendly than what we used would likely yield better results.

---

### Decision · Program_Chairs · 2019-12-19

**Decision:**

Accept (Talk)

**Comment:**

The reviewers develop a novel technique for training neural networks that are provably robust to adversarial attacks, by combining provable defenses using convex relaxations with latent adversarial attacks that lie in the gap between the convex relaxation and the true realizable set of activations at a layer of the network. The authors show that the resulting procedure is computationally efficient and able to train neural networks to attain SOTA provable robustness to adversarial attacks.

The paper is well written and clearly explains an interesting idea, backed by thorough experiments. The reviewers were in consensus on acceptance and relatively minor concerns were clearly addressed in the rebuttal phase.

Hence, I strongly recommend acceptance.